# Antimicrobial Activities of LL-37 Fragment Mutant-Poly (Lactic-Co-Glycolic) Acid Conjugate against *Staphylococcus aureus*, *Escherichia coli*, and *Candida albicans*

**DOI:** 10.3390/ijms22105097

**Published:** 2021-05-12

**Authors:** Takeshi Mori, Miyako Yoshida, Mai Hazekawa, Daisuke Ishibashi, Yoshiro Hatanaka, Toshihiro Nagao, Rie Kakehashi, Honami Kojima, Rio Uno, Minoru Ozeki, Ikuo Kawasaki, Taku Yamashita, Junichi Nishikawa, Takahiro Uchida

**Affiliations:** 1Faculty of Pharmaceutical Sciences, Mukogawa Women’s University, 11-68 Koshien 9-Bancho, Nishinomiya City 663-8179, Japan; mw319016@mukogawa-u.ac.jp (T.M.); miyakoy@mukogawa-u.ac.jp (M.Y.); h_kojima@mukogawa-u.ac.jp (H.K.); unor@mukogawa-u.ac.jp (R.U.); ozekim@mukogawa-u.ac.jp (M.O.); ikuo_k@mukogawa-u.ac.jp (I.K.); taku@mukogawa-u.ac.jp (T.Y.); nisikawa@mukogawa-u.ac.jp (J.N.); 2Department of Immunological and Molecular Pharmacology, Faculty of Pharmaceutical Science, Fukuoka University, 8-19-1 Nanakuma, Jonan-ku, Fukuoka City 814-0180, Japan; mhaze@fukuoka-u.ac.jp (M.H.); dishi@fukuoka-u.ac.jp (D.I.); 3Osaka Research Institute of Industrial Science and Technology, 1-6-50 Morinomiya, Joto-ku, Osaka City 536-8553, Japan; hatanaka@omtri.or.jp (Y.H.); nagao@omtri.or.jp (T.N.); rie@omtri.or.jp (R.K.)

**Keywords:** antimicrobial peptide, mutant peptide, conjugation with poly (lactic-co-glycolic) acid, scanning electron microscopy, transmission electron microscopy

## Abstract

Various peptides and their derivatives have been reported to exhibit antimicrobial activities. Although these activities have been examined against microorganisms, novel methods have recently emerged for conjugation of the biomaterials to improve their activities. Here, we prepared CKR12-PLGA, in which CKR12 (a mutated fragment of human cathelicidin peptide, LL-37) was conjugated with poly (lactic-co-glycolic) acid (PLGA), and compared the antimicrobial and antifungal activities of the conjugated peptide with those of FK13 (a small fragment of LL-37) and CKR12 alone. The prepared CKR12-PLGA was characterized by dynamic light scattering and measurement of the zeta potential, critical micellar concentration, and antimicrobial activities of the fragments and conjugate. Although CKR12 showed higher antibacterial activities than FK13 against *Staphylococcus aureus* and *Escherichia coli*, the antifungal activity of CKR12 was lower than that of FK13. CKR12-PLGA showed higher antibacterial activities against *S. aureus* and *E. coli* and higher antifungal activity against *Candida albicans* compared to those of FK13. Additionally, CKR12-PLGA showed no hemolytic activity in erythrocytes, and scanning and transmission electron microscopy suggested that CKR12-PLGA killed and disrupted the surface structure of microbial cells. Conjugation of antimicrobial peptide fragment analogues was a successful approach for obtaining increased microbial activity with minimized cytotoxicity.

## 1. Introduction

Antimicrobial peptides (AMPs), as host defense peptides, show potential as new therapeutic classes of antimicrobials [1]. These peptides exert biological activities by interacting with the plasma membrane to disrupt the membrane and lyse the cell, or are taken up by the target cell depending on the amino acid composition of the peptide [2]. In general, AMPs have an amphipathic and cationic structure with a positive net charge from +3 to +9 and range in size from 12 to 50 amino acid residues [3,4,5,6]. Another desirable property of AMPs is their broad-spectrum activity, enabling the targeting of several types of pathogenic microorganisms.

Derivatization of AMPs is a particularly successful approach for increasing antimicrobial activity. A hybrid peptide designed by combining the α-helix fragment with the core antimicrobial cationic fragment was reported to have higher antibacterial activity than either fragment alone [7]. The net positive charge and helical content have been suggested as important contributors to the antibacterial activity of AMPs.

LL-37, an amphipathic α-helical peptide isolated from humans and belonging to the cathelicidin family [8], is a cationic peptide consisting of 37 amino acids and has been reported to exhibit antimicrobial activity [9,10,11]. Human LL-37 and its shorter mutant composed of the bactericidal core peptides may exert antimicrobial activities while being cost-effective to produce [12,13].

Poly (lactic-co-glycolic) acid (PLGA) has been used as a high-hydrophobic biomaterial because of its biodegradability and potential to control the timing of drug release from nanoparticles [14,15]. However, PLGA shows substantial hydrophobicity and relatively poor drug-loading capacity; therefore, studies have been conducted to further improve its physicochemical characteristics via conjugation with other polymers and amino acids [16].

In this work, we evaluated the mechanism of an AMP-PLGA conjugate and attempted to enhance its antimicrobial activity. We focused on FK13, a fragment of LL-37, and its mutant, CKR12. Moreover, CKR12 was conjugated with PLGA as CKR12-PLGA, and the conjugate and fragments were evaluated to determine their antimicrobial activity against *Staphylococcus aureus*, *Escherichia coli*, and *Candida albicans*. This conjugate may be applied to deliver antibacterial drugs based on its ability to self-assemble into a micellar structure.

## 2. Results

### 2.1. Predicted Structural Parameters of Peptides

We first evaluated the structural properties of the peptides. Figure 1 shows the amino acid sequences and secondary structure predictions of LL-37, FK13, and CKR12.

The prediction suggested that the fragments FK13 and CKR12 had higher helical contents (84.6%) compared to LL-37 (78.4%). The molecular properties were also calculated, and the values are summarized along with the helical contents in Table 1.

As shown in Table 1, helical content appeared to correlate with the percentage of hydrophobic residues, whereas the net charges were altered by the mutations in CKR12. CKR12 showed the highest net charge among the studied peptides, with a value +3 higher than that of FK13. Additionally, the net charge of CKR12 was close to that of the parent peptide LL-37 (+6).

### 2.2. Characterization of CKR12 and Its Conjugate, CKR12-PLGA

Subsequently, the designed peptide CKR12 was prepared and conjugated with PLGA. Conjugation was performed according to a previous report [17] with slight modifications, as shown in Figure 2.

To confirm that conjugation was successful, we measured the Fourier transform infrared (FT-IR) spectra of PLGA and CKR12-PLGA. Both spectra showed comparable bands corresponding to the -OH and -COOH groups of PLGA. ^1^H-nuclear magnetic resonance (NMR) spectroscopy was also performed for PLGA and the conjugate. The chemical shifts of the signals assigned to PLGA were indistinguishable among the spectra. The FT-IR and NMR results indicate that the prepared CKR12-PLGA contained PLGA as either a conjugation or mixture.

To clarify whether CKR12 and PLGA were conjugated, we measured the critical micelle concentration (CMC), dynamic light scattering (DLS) spectra, and zeta potential of the prepared CKR12-PLGA (Figure 3).

The relative intensities of scattered light were monitored and traced as the mean diameter of the polymeric conjugate, which was found to be 217.3 ± 30.4 nm and showed monophasic aspects (Figure 3A). The zeta potential was 39.6 ± 4.4 mV (Figure 3B). These results suggest that the synthesized conjugate formed hydrophilic micelles with an estimated CMC of 12 µM based on monitoring of the absorbance changes (data not shown). These values are discussed in more detail below.

### 2.3. Antimicrobial and Hemolytic Activities

After confirming the conjugation of CKR12 and PLGA, we examined the antibacterial activities of the fragments and conjugate using the micro-broth dilution method (Table 2).

The MIC values in each microorganism were statistically evaluated to estimate the *p*-value; however, this value could not be determined for *S. aureus* because of the statistical dispersion of the results. First, the MIC values for FK13 were determined as 14.1 µM for *S. aureus* and *E. coli*, and 28.1 µM for *C. albicans*. Comparison of the MIC values for FK13 and CKR12-PLGA showed that all three microorganisms, *S. aureus*, *E. coli*, and *C. albicans*, were more susceptible to CKR12-PLGA than to FK13 (*E. coli*, *p* < 0.01; *C. albicans*, *p* < 0.05). For example, *C. albicans* showed 24.2 µM for the conjugate, and the value was slightly lower than that of FK13 (28.1 µM). Two microorganisms, *S. aureus* and *E. coli*, were more susceptible to CKR12 than to CKR12-PLGA (*E. coli*, *p* < 0.01). The effects of PLGA alone was examined and confirmed showing minimal antimicrobial activity against the three studied microorganisms. Moreover, no hemolytic activity was observed even at the highest concentration (232 µM) of CKR12-PLGA (data not shown).

### 2.4. Scanning and Transmission Electron Microscopic Imaging

We next monitored whether CKR12-PLGA affected the morphology of *S. aureus*, *E. coli*, and *C. albicans*. As shown in Figure 4, the three microorganisms were evaluated by scanning electron microscopy (SEM) with (right panel) or without (left panel) the conjugate.

*Staphylococcus aureus* was monitored by SEM, as shown in Figure 4A. Although spherical aspects were observed regardless of the presence of CKR12-PLGA, the surfaces of the microorganisms were slightly coarse following treatment with the conjugate. These effects were clearly observed for *E. coli* (Figure 4B), which showed several pits with deformation. Moreover, *C. albicans* had a deformed and wrinkled surface in the presence of CKR12-PLGA (Figure 4C).

We next performed transmission electron microscopy (TEM) of *S. aureus, E. coli,* and *C. albicans* cells in the absence (left panel) and presence (right panel) of CKR12-PLGA (Figure 5).

TEM was performed to directly observe the surface and interior of the microorganisms. All three untreated microorganisms showed dark and smooth surfaces. The surface of *S*. *aureus* showed no morphological changes after treatment with CKR12-PLGA, and the results were similar to those observed by SEM, although a crack was observed in the inner part of the cell (Figure 5A, right panel). Both *E. coli* and *C. albicans* cells showed deformed surfaces. In addition, the inner parts were spongy, which may have caused the deformation of both cells (Figure 5B,C, right panel). This effect was not a contradiction among the electron microscopic measurements.

## 3. Discussion

This study was conducted to investigate the antimicrobial activity of AMP analogs with broad-spectrum activities. Structure-activity modeling of the structural features responsible for the activity of tethered peptides have indicated that the extent and positioning of positive charges and hydrophobic residues influences AMP activity [18]. The positive charge is thought to be closely related to the electrostatic binding force between the AMP and the negatively charged bacterial cell membrane [19,20], and the helical structure may enable deep insertion of the AMP into the bacterial cell membrane [21,22]. Our study also supports the helical structure’s role in cell penetration. Remarkably, the peptide with the tagging cysteine residue at the N-terminus showed a greater helical structure content (85%) compared to having it at the C-terminus (77%), although the positive charge value did not differ. Thus, AMP fragment analogs show high antimicrobial activity, and these functions depend on structural and physicochemical properties [7].

The conjugate, CKR12-PLGA, was next assessed by DLS and zeta potential measurements. As shown in Figure 3, both results showed uniform dispersions and suggested that the self-assembled micelles were of similar sizes. These results were confirmed by measuring the CMC values. Hence, our AMP fragment analogs were an appropriate size to form micelles and exert antimicrobial activities.

Comparison of the antibacterial activities of FK13, CKR12, and CKR12-PLGA and MIC of CKR12 revealed lower antibacterial activity against *S. aureus* and *E. coli* compared to those of FK13. Based on the MIC value, the antifungal activity of CKR12 was higher than that of FK13. The MIC values of CKR12-PLGA showed lower antibacterial activity against *S. aureus* and *E. coli* (MIC < CMC) and lower antifungal activity against *C. albicans* (MIC > CMC) compared to those of FK13. AMPs rapidly neutralize a broad range of microbes, including both gram-positive and gram-negative bacteria, mainly by binding to and perturbing the cell envelopes containing lipoteichoic acid and lipopolysaccharide, respectively, as well as their cytoplasmic membranes [23,24]. In fact, the greater CMC values compared to the MIC values suggests that the AMP analog can affect microorganisms as a micellar structure at effective concentrations. Additionally, the CMC value may be an important factor in improving drug delivery, which will improve antibacterial activity [25]. Each MIC value was lower than the CMC, indicating that the cationic peptide part of CKR12-PLGA attacked the bacterial cell membrane of *S. aureus* and *E. coli.* The surface charge of *C. albicans* has a zeta potential of approximately −20 to −35 mV, which is less than that of the former two bacteria, and shows high lipophilicity [26] despite the hydrophobic properties of the biodegradable polymer PLGA [14]. The MIC value was higher than the CMC for *C. albicans*, suggesting that the self-assembled micelle of CKR12-PLGA can attack the bacterial cell membrane of *C. albicans*, as shown in Figure 6.

Deformation of the microorganisms was observed in the presence of CKR12-PLGA; however, the morphological changes of *S. aureus* differed from those of *E. coli* and *C. albicans*. Previous reports suggested that the antimicrobial effects of AMPs are conferred by their α-helical structures, which can form pores in the membrane [10,27]. Our study suggests that the layer composed of teichoic acids protected *S. aureus*, and that CKR12-PLGA was less effective for deforming the cell surface structure of *S. aureus* but still gave a certain warp by forming pores. Thus, the antimicrobial properties of CKR12-PLGA observed by electron microscopy agreed with the measured antimicrobial activities.

In this study, the antimicrobial activities of the LL-37 fragment mutant (CKR12) against *S. aureus* and *E. coli* and those of a mutant conjugated with PLGA (CKR12-PLGA) against *S. aureus*, *E. coli*, and *C. albicans* were evaluated. Conjugation of the AMP fragment analogues and biopolymers was a successful approach for obtaining increased antimicrobial activity and achieving a broad antibacterial spectrum with no hemolytic activity, although the mechanism has not been fully clarified yet. Further studies are needed to improve the formation of the micellar structure of AMP analog conjugates with PLGA, which will be a useful tool for delivering antimicrobial drugs.

## 4. Materials and Methods

### 4.1. Materials

The LL-37 fragment (17–29) was purchased from Funakoshi Co., Ltd. (Tokyo, Japan). The mutant peptide (CKRIVKRIVKKWLR) was synthesized by the Toray Research Center (Tokyo, Japan). Poly (d,l-lactic-co-glycolic acid) (PLGA7510, MW: 10,000), 3-(2-pyridyldithio) propionylhydrazide (PDPH), *N*,*N*′-dicyclohexylcarbodiimide, *N*-hydroxysuccinimide, and Sudan III were purchased from Wako Pure Chemical Industries (Osaka, Japan).

Three microbial strains were purchased from the Biological Resource Center, National Institute of Technology and Evaluation (NBRC) (Tokyo, Japan) and Japan Collection of Microorganisms in Riken BioResource Research Center (Ibaragi, Japan). The strains used in this study were *S. aureus* (NRBC 12732), *E. coli* (NRBC 3972), and *C. albicans* (JCM 1542).

### 4.2. Biological Information Analysis of LL-37, LL-37 Fragment (17–29), and Mutant Peptide

The primary structure characteristics (including net charge and hydrophobic residues) of LL-37, LL-37 fragment (17–29) and the mutant peptide were predicted using the Antimicrobial Peptide Database (http://aps.unmc.edu/AP/main.php, accessed date: 10 September 2020) and NetSurfP-2.0 [28].

### 4.3. Preparation of PLGA Conjugate

The CKR12-PLGA hybrid block copolymer was synthesized as described previously [29] with minor modifications. CKR12 was conjugated to PLGA-PDPH via a disulfide exchange reaction [17]. The synthesized CKR12-PLGA hybrid conjugates were predicted to form self-assembled micelles in aqueous solutions, resulting in a substantially increased charge density of clustered CKR12 in the outer shell.

### 4.4. Structural Characterization by ^1^H-NMR and FT-IR Spectroscopy

^1^H-NMR spectra of the conjugate and PLGA were measured using CDCl_3_ as the solvent and tetramethylsilane as the standard, with an ECP-500 NMR spectrometer (Jeol, Tokyo, Japan) at a frequency of 500 MHz. FT-IR can be used to analyze and compare changes in polymer binding to drugs and characterize protein structures. The infrared absorption spectra of the PLGA conjugate and PLGA were measured in CHCl_3_ with an IRAffinity-1 (Shimadzu Corporation, Kyoto, Japan). Both spectra were recorded at room temperature.

### 4.5. Measurement of CMC

The CMC is defined as the lowest concentration of surfactant at which micelles form in water. The CMC was determined from the absorbance of Sudan III at the maximum absorption wavelength (511 nm) by the dye solubilization method using a spectrophotometer [30].

### 4.6. Particle Size Distribution and Surface Charge Measurement of PLGA-Conjugated Micelles

The sizes and surface charges of PLGA-conjugated micelles were measured using a DLS instrument (Zetasizer; Malvern Instruments, Ltd., Malvern, UK). PLGA-conjugated micelles were dissolved in distilled water. The effective hydrodynamic diameters and zeta potentials of the PLGA-conjugated micelles were measured in triplicate.

### 4.7. Hemolytic Assay for CKR12-PLGA

The CKR12-PLGA concentrations (from 0.06 to 232 µM) at which hemolysis of sheep erythrocytes occurred was measured at 540 nm. Sheep erythrocytes in 10 mM phosphate-buffered saline (PBS)/10% (*v*/*v*) Triton X-100 (Sigma, St. Louis, MO, USA) were used as negative and positive controls, respectively.

### 4.8. Antimicrobial Assays

The antibacterial activity of the peptides was characterized by determining their MICs. Bacterial strains (*S. aureus* and *E. coli*) in the exponential growth phase were diluted to a concentration of 4 × 10^4^ CFU/mL with Mueller-Hinton broth and then diluted to 4 × 10^4^ CFU/mL with YM broth; 180 µL of the culture was dispensed into each well of a 96-well microtiter plate. Susceptibility tests were performed using two-fold standard broth microdilutions of FK13, CKR12, and CKR12-PLGA according to Clinical and Laboratory Standards Institute guidelines. MICs were measured using test samples from 232 to 0.06 µM. Each analysis was performed in triplicate. The lowest concentration (highest dilution) required to prevent the growth of microorganisms was regarded as the MIC.

### 4.9. Imaging by SEM and TEM

For microscopic measurements, the three microorganisms were prepared as follows. *Escherichia coli* and *S. aureus* cell suspensions (1 × 10^8^ CFU/mL) were incubated with CKR12-PLGA (61 µM) at 37 °C with shaking at 180 rpm. *Candida albicans* cell suspensions (1 × 10^8^ CFU/mL) were incubated with CKR12-PLGA (61 µM) at 27 °C. Two milliliters of the mixture were collected at 2 h and centrifuged at 1000× *g* for 5 min, and the supernatant was discarded. The bacterial cells were washed with PBS three times, fixed with 2.5% glutaraldehyde at 4 °C overnight, washed twice with PBS again, and postfixed for 2 h with 1% OsO_4_. The prepared bacterial pellets were dehydrated with a graded series of ethanol, following which the dried cells were coated with gold and observed using a scanning electron microscope (JSM-7800, Jeol). The prepared pellets were observed using a transmission electron microscope (JEM-2100, Jeol) for imaging.

### 4.10. Statistical Analyses

Experimental data were analyzed by one-way analysis of variance using the general linear model procedure of BellCurve for Excel^®^. Statistically significant effects were further analyzed, and means were compared by the Tukey-Kramer test. Statistical significance was set at *p* < 0.05.

## Figures and Tables

**Figure 1 ijms-22-05097-f001:**
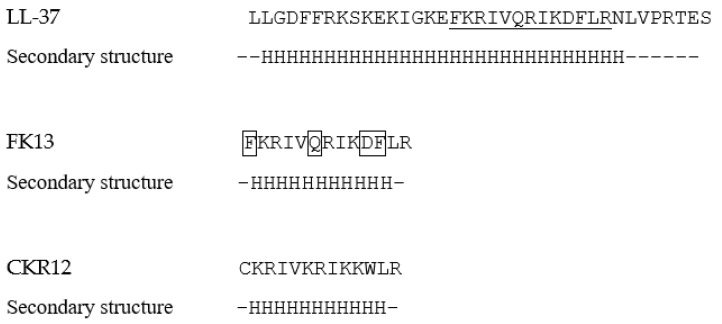
Predicted structures of LL-37, FK13, and CKR12. Fragmented sequence in LL-37 is underlined, and mutated residues in CKR12 are enclosed in the sequence of FK13. The predicted secondary structures are represented underneath the sequences as “H” for helical and “-” for random coiled structures.

**Figure 2 ijms-22-05097-f002:**
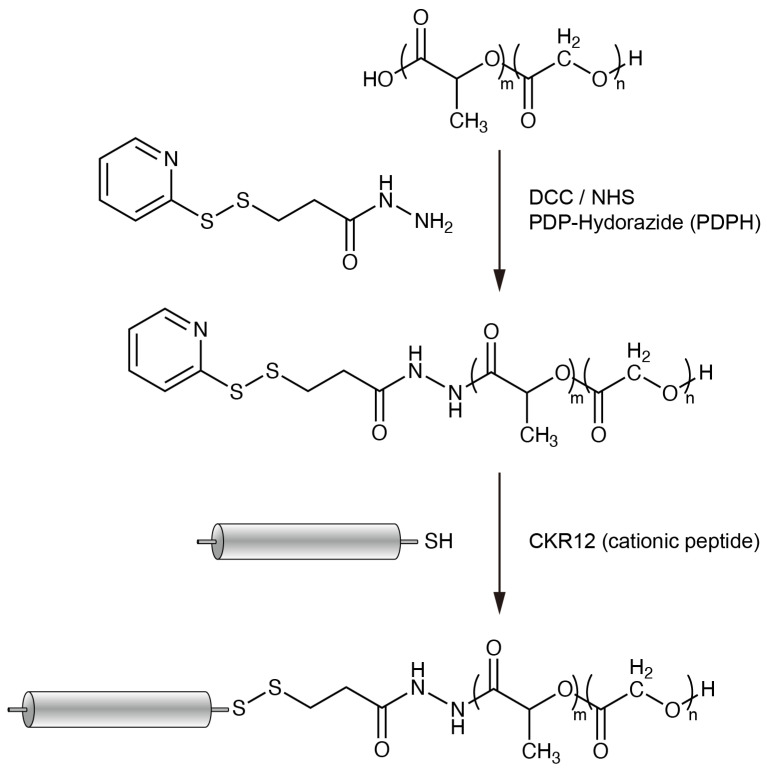
Summary of synthesis of CKR12-PLGA conjugate via a cleavable disulfide linkage. CKR12 is shown as a cylinder.

**Figure 3 ijms-22-05097-f003:**
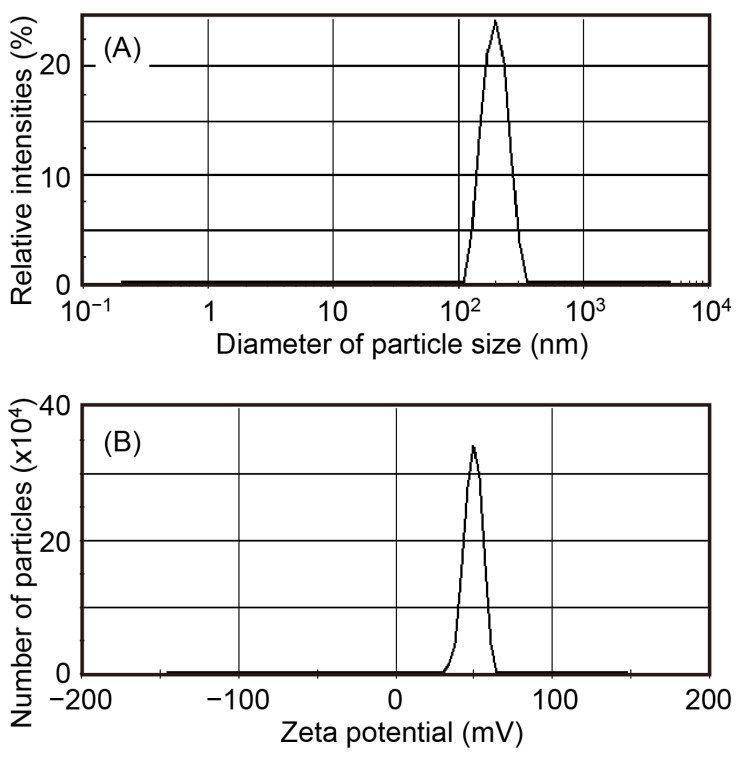
Particle size (**A**) and zeta potential values (**B**) of CKR12-PLGA conjugate micelle.

**Figure 4 ijms-22-05097-f004:**
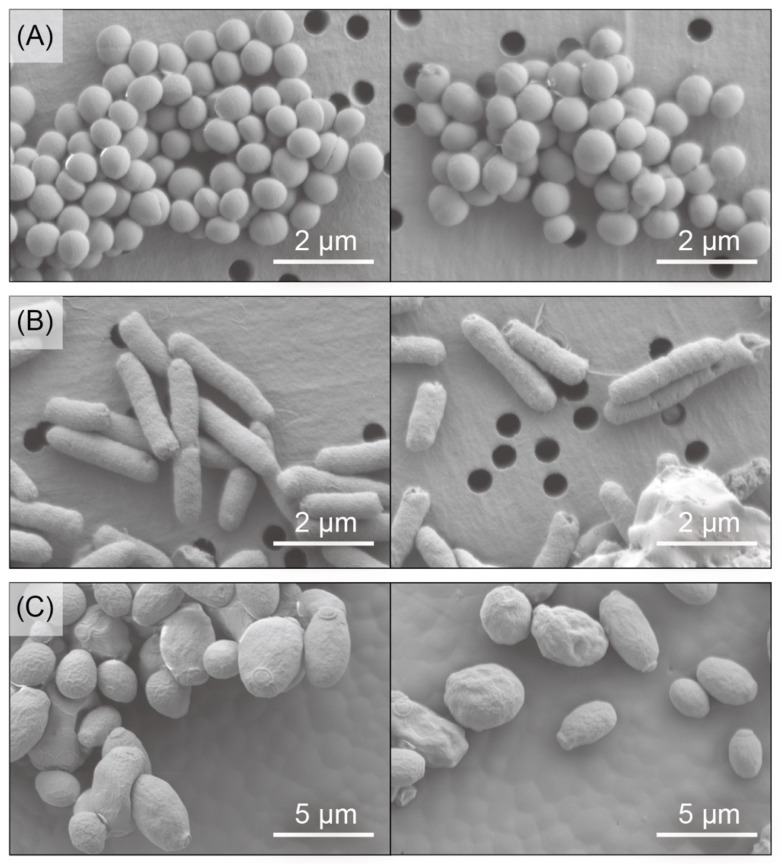
Scanning electron microscopic images of *S. aureus* (**A**), *E. coli* (**B**), and *C. albicans* (**C**) cells in the absence (left panel) and presence (right panel) of 110 µM CKR12-PLGA.

**Figure 5 ijms-22-05097-f005:**
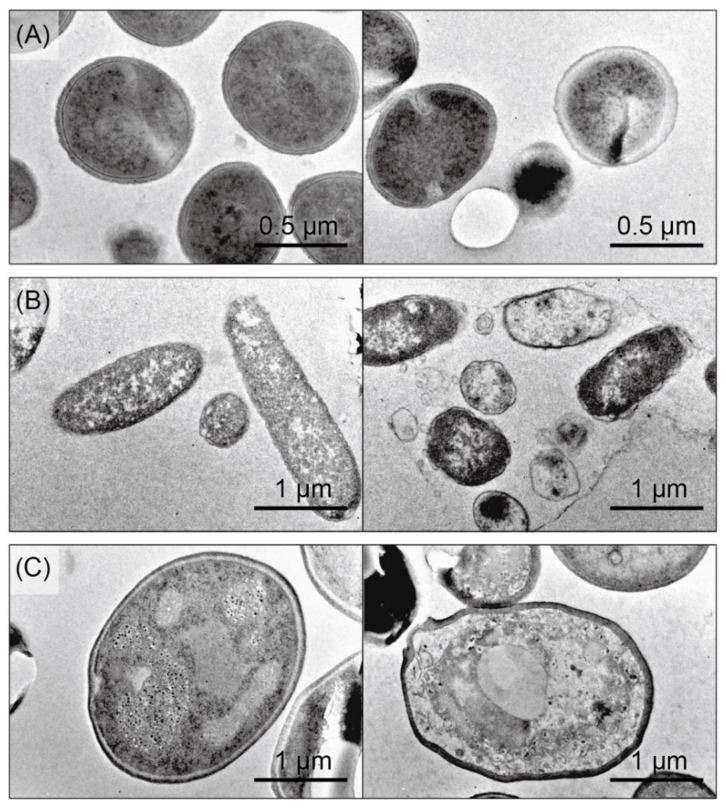
Transmission electron microscope images of *S. aureus* (**A**), *E. coli* (**B**), and *C. albicans* (**C**) cells. The microorganisms are shown in the absence (left panel) and presence (right panel) of 61.3 µM CKR12-PLGA.

**Figure 6 ijms-22-05097-f006:**
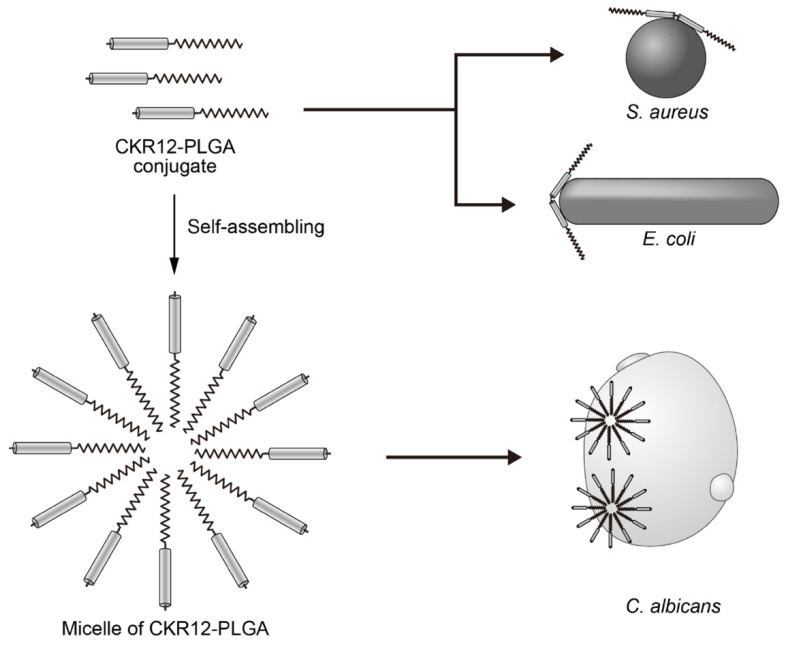
Schematic mechanisms of individual and self-assembled micelles of CKR12-PLGA against *S. aureus*, *E. coli*, and *C. albicans*.

**Table 1 ijms-22-05097-t001:** Expected structural parameters of LL-37, FK13, and CKR12.

Peptide	Helical Content (%)	Hydrophobic Residues (%)	Net Charge
LL-37	78.4	35	+6
FK13	84.6	46	+4
CKR12	84.6	46	+7

**Table 2 ijms-22-05097-t002:** Minimum inhibitory concentration (MIC) of FK13, CKR12, and CKR12-PLGA against microorganisms. The MIC values (μM) were determined as the average values obtained from three experiments.

Organism	MIC Value (µM)				*p*-Value
	FK13	CKR12	CKR12-PLGA *	PLGA	
*S. aureus* (NBRC 12732)	14.1 ± 0.00	0.91 ± 0.00	3.63 ± 0.00	362 ± 0.00	
*E. coli* (NBRC 3972)	14.1 ± 0.00 ^b^	2.47 ± 1.13 ^d^	9.67 ± 1.19 ^c^	362 ± 0.00 ^a^	<0.01
*C. albicans* (JCM 1542)	28.1 ± 0.00 ^c^	58.0 ± 0.00 ^b^	24.2 ± 8.37 ^d^	181 ± 0.00 ^a^	<0.01

* MIC of the conjugate is stated in terms of the equivalent of the antimicrobial peptide fragment analogue equivalent. ^a,b,c^ Means with different superscripts within the same row differ significantly (*p* < 0.05).

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
