# Peer review of "Antimicrobial Activities of LL-37 Fragment Mutant-Poly (Lactic-Co-Glycolic) Acid Conjugate against Staphylococcus aureus, Escherichia coli, and Candida albicans"

_ijms, 2021, doi:10.3390/ijms22105097_

Round 1

Reviewer 1 Report

The subject of the article is important and actual. 

I have the following comments and questions for the authors. There are many awkward phrases that I do not point out here; I only point out those where the meaning cannot be interpreted:

The paragraph between 165-177 is not clear can be rewrite in more clear format.

The conclusion need to clear and specific. My recommendation is to focus on short conclusion.

Please recheck the References order.

Please double check the article by a native English reader.

Thanks again for the chance of reading the article.

Author Response

Reviewer #1

Comments and Suggestions for Authors:

The subject of the article is important and actual.

I have the following comments and questions for the authors. There are many awkward phrases that I do not point out here; I only point out those where the meaning cannot be interpreted:

Thank you for your suggestion. According to your comment, our manuscript has been rechecked thoroughly and edited carefully by a native speaker in Editage providing proofreading services. Our revised phrases and sentences were highlighted in yellow in the manuscript.

Besides, we replied your pointed comments as the following.

1) The paragraph between 165-177 is not clear can be rewrite in more clear format.

According to your comment, we added several sentences to make clear for comparison of the values. Also, several modifications were performed to improve our explanation.

2) The conclusion need to clear and specific. My recommendation is to focus on short conclusion.

To make clear and specific, we modified the conclusion slightly.

3) Please recheck the References order.

Thank you very much for your suggestion. Although our rechecks suggested that the references were surely lined up in order, we apologize and added lacking abbreviation marks "." to references (No. 2, 15, 17, 25, and 30) and added lacking page number to references (No. 1, 2, 4, 5, 8, 9, 10, 11, 12, 14, 15, 16, 17, 19, 21, 22, 24, 26, 27 and 30).

4) Please double check the article by a native English reader.

The manuscript has been checked thoroughly by a native English reader which provided an English editing company, Editage.

Reviewer 2 Report

The manuscript “Antimicrobial Activities of LL37 Fragment Mutant-Poly (Lactic-Co-Glycolic) Acid Conjugate Against Staphylococcus aureus, Escherichia coli, and Candida albicans analyses the mechanism of an AMP-PLGA [antimicrobial peptides-poly(lactic-co-glycolic)acid] conjugate and attempted to enhance its antimicrobic activiy.

The results showed that conjugation of antimicrobial peptide fragment analogs was an effective approach to achieve increased microbial activity with minimized cytotoxicity.

In my opinion this work is very interesting, well structured and the methods used are correct.

There are no comments.

Author Response

Reviewer #2

Comments and Suggestions for Authors:

The manuscript “Antimicrobial Activities of LL37 Fragment Mutant-Poly (Lactic-Co-Glycolic) Acid Conjugate Against Staphylococcus aureus, Escherichia coli, and Candida albicans” analyses the mechanism of an AMP-PLGA [antimicrobial peptides-poly(lactic-co-glycolic)acid] conjugate and attempted to enhance its antimicrobic activity.

The results showed that conjugation of antimicrobial peptide fragment analogs was an effective approach to achieve increased microbial activity with minimized cytotoxicity.

In my opinion this work is very interesting, well structured and the methods used are correct.

There are no comments.

We appreciate your encouraged comment and acceptance for publication.